# A Contribution to the Phylogeny and Taxonomy of *Hydnum* (Cantharellales, Basidiomycota) from China

**DOI:** 10.3390/jof10020098

**Published:** 2024-01-25

**Authors:** Ming Zhang, Chaoqun Wang, Hongfen Bai, Wangqiu Deng

**Affiliations:** 1Guangdong Provincial Key Laboratory of Microbial Culture Collection and Application, State Key Laboratory of Applied Microbiology Southern China, Institute of Microbiology, Guangdong Academy of Sciences, Guangzhou 510070, China; dayangtutu@163.com; 2Chuxiong Yi Autonomous Prefecture Forestry and Grassland Science Research Institute, Chuxiong 675000, China; cxbhf111@163.com

**Keywords:** Hydnaceae, molecular systematics, morphology, new record, new species

## Abstract

*Hydnum* is a well-characterized genus in the family Hydnaceae of Cantharellales and is characterized by spinose hymenophores. In this study, an ITS phylogenetic overview and a multilocus (ITS-nrLSU-*tef*1) phylogenetic tree of *Hydnum* were carried out. On the basis of morphological characteristics and phylogenetic results, seven species from China were confirmed, described, illustrated, and compared with similar species, including three new species, i.e., *H. longipes*, *H. microcarpum*, and *H. sinorepandum*, and four known species, i.e., *H. cremeoalbum*, *H. melitosarxm*, *H. orientalbidum*, and *H. pinicola* were recorded for the first time in China. A key to the species of *Hydnum* in China was provided.

## 1. Introduction

*Hydnum* L., typified by *H. repandum* L., belongs to Basidiomycota, Basidiomycetes, Cantharellales, Hydnaceae [1,2]. Morphologically, *Hydnum* species are notable for having spinose hymenophores; in addition, they have pileate and stipitate basidiocarps, stichic basidia, and smooth basidiospores. Phylogenetically, *Hydnum* is the sister group of *Sistotrema*, and they are closely related to the genera *Cantharellus* and *Craterellus* [2]. Economically, some *Hydnum* species (named “hedgehogs”, “sweet tooth”, or “tooth fungus”) are collected and consumed as edible mushrooms. For example, *H. jussii* Niskanen, Liimat. & Kytöv, “*H. repandum* complex”, *H. vesterholtii* Olariaga, Grebenc, Salcedo & M.P. Martín are important wild edible mushrooms in southwestern China [3,4]. Ecologically, as a cosmopolitan genus, *Hydnum* plays an important role in forest ecology and can form ectomycorrhizal associations with many trees, e.g., Betulaceae, Fagales, and Pinaceae [2,5,6].

Phylogenetic studies proved that *Hydnum* is a monophyletic group [1,2,7], and five subgenera, i.e., subg. *Alba*, subg. *Brevispina*, subg. *Hydnum*, subg. *Pallida*, and subg. *Rufescentia* have been recognized within the genus [2,6]. Apart from subg. *Alba*, the other four subgenera are monophyletic groups. Of the five subgenera, subg. *Rufescentia* contains the most known species.

Studies on the species diversity of *Hydnum* in China have accelerated in recent years. Previously, only three species (*H. repandum*, *H. repandum* var. *album*, and *H. rufescens*) were recorded in China before 2016 [5,8,9]. Feng et al. [5] conducted a molecular phylogenetic study on *Hydnum* in China, and the results showed that *Hydnum* species are very diverse in China. Cao et al. [2] reported twelve species of *Hydnum* in China, including ten new species. Until now, about 18 species have been reported from China [2,3,4,5,8,9,10]. In recent years, we have carried out investigations on the species diversity of Hydnaceae in southern China and found 11 new species of *Cantharellus* [11,12]. In this study, three new species of *Hydnum* are described and discussed, and four species are described in China for the first time.

## 2. Materials and Methods

### 2.1. Morphological Studies

Photographs and field notes of the fresh basidiomata were taken in the field or laboratory. Specimens were dried and deposited in the Fungarium of Guangdong Institute of Microbiology (GDGM). Descriptions of macro-morphological characteristics and habitats were obtained from field notes and photographs. Color codes followed Kornerup and Wanscher [13]. Micro-morphological observations were carried out on tissues stained with 5% KOH and 1% aqueous Congo red under a light microscope (Olympus BX51, Tokyo, Japan). For basidiospore size descriptions, the notation (a–)b–c(–d) is used, where the range b–c represented 90% or more of the measured values and ‘a’ and ‘d’ represented the extreme values; ‘av.’ represents the mean range of basidiospore length or width; Q referred to the length/width ratio of an individual basidiospore; and Q_m_ referred to the average Q value of all basidiospores ± sample standard deviation. All line drawings of microstructures were made based on rehydrated materials.

### 2.2. DNA Extraction, PCR Amplification and Sequencing

Total genomic DNA of the voucher specimens was extracted using the Sangon Fungus Genomic DNA Extraction kit (Sangon Biotech Co. Ltd., Shanghai, China), according to the manufacturer’s instructions. Primer pairs ITS5/ITS4 [14], LR0R/LR5 [15], and tef1F/tef1R [16] were used for amplifying and sequencing these sequences: ITS, nrLSU, and *tef*1, respectively. Polymerase chain reaction (PCR) reactions were performed in a total volume of 25 μL containing 0.5 μL template DNA, 11 μL distilled water, 0.5 μL of each primer, and 12.5 μL 2 × PCR mix (DreamTaqtm Green PCR Master Mix, Fermentas, MA, USA). PCR amplification reactions were performed in a Professional Standard Thermocycler (Biometra, Göttingen, Germany) as follows: denaturation at 95 °C for 4 min; 35 cycles (denaturation at 94 °C for 60 s, annealing at 53 °C (ITS and LSU)/50 °C (*tef*1) for 60 s, and extension at 72 °C for 60 s); and a final extension at 72 °C for 8 min. The PCR products were electrophoresed on 1% agarose gels and sequencing was performed on an ABI Prism^®^ 3730 Genetic Analyzer (PE Applied Biosystems, Foster, CA, USA) at the Beijing Genomic Institute (BGI). The raw sequences were assembled and edited using SeqMan implemented in Lasergene v7.1 (DNASTAR Inc., Madison, WI, USA) and deposited in GenBank.

### 2.3. Phylogenetic Analyses

Sequences generated in this study and those downloaded from GenBank were combined and used for phylogenetic reconstruction. The sequence matrix of ITS, nrLSU, and *tef*1 were separately aligned with MAFFT v7 software using the E-INS-i strategy [17] and manually adjusted in MEGA 6 [18]. Phylogenetic analyses were performed in PhyloSuite [19]. Maximum likelihood phylogenies were inferred using IQ-TREE [20]. Best models for ITS datasets were searched via ModelFinder [21], and the ITS-LSU-*tef*1 combined datasets were searched via PartitionFinder 2 [22]. The phylogenetic trees were visualized using FigTree v1.4.23.

## 3. Results

### 3.1. Molecular Phylogeny

In the ITS dataset, 410 sequences were included for phylogenetic analysis. The alignment had 843 characters including gaps, and K80 + G4 was selected as the best model for maximum likelihood phylogenies. *Sistotrema muscicola* (Pers.) S. Lundell was selected as an outgroup based on recent studies [2,6]. The ITS phylogenetic tree is shown in Figure 1. In the concatenated (ITS + LSU + *tef*1) dataset, the alignment has 274 samples with 3348 columns, 1224 distinct patterns, 764 parsimony-informative, 160 singleton sites, and 2424 constant sites. The best-fit models HKY + F + R3 and TIM3 + F + R2 were selected for the ITS + *tef*1 and LSU regions, respectively. The multilocus (ITS + LSU + *tef*1) phylogenetic tree is shown in Figure 2. In total, 51 sequences were newly generated and deposited in GenBank in present study.

Phylogenetic analyses show that the maximum likelihood topologies obtained from the ITS and the concatenated (ITS + LSU + *tef*1) datasets in our study are similar to the phylogram in previous studies [2,23]. The newly generated sequences were classified into seven species-level clades, all with high support values (BS = 100%). Among these, GDGM87902 and GDGM87902-1 formed a distinct lineage in the subg. *Brevispina*; GDGM82458, together with an unnamed sequence (*Hydnum* sp. HKAS57385), formed a monophyletic lineage which nested into the subg. *Rufescentia*; GDGM82445, GDGM82416, GDGM82382, and GDGM90134, together with five sequences named as *Hydnum* sp. 8, formed a distinct lineage in subg. *Hydnum*; GDGM93020 and GDGM83047 nested into the *H. pinicola* clade; GDGM93011 and GDGM87013 nested into the *H. cremeoalbum* clade; and GDGM93480, GDGM93019, and GDGM91301 nested into the *H. orientalbidum* clade.

### 3.2. Taxonomy

Below, three new species and four known species of *Hydnum* from China are described, illustrated, and discussed. The descriptions are arranged alphabetically. 

*Hydnum cremeoalbum* Liimat. & Niskanen; Figure 3a and Figure 4.

Basidiocarps medium to large sized, solitary to scattered. Pileus 30–100 mm broad, convex to subapplanate, usually depressed in the center, margin thin, even, decurved at first; surface dry, glabrous, white to cream white, unchanging when bruised. Context 3–8 mm thick, white, unchanging when exposed. Hymenophore spinose, adnate, not decurrent; spines conical to aciculate, crowded, subulate, 2–5 mm long, shortest near the pileus margin, yellowish-white, orange-white to pale orange (3A2–5A2, 4A3–5A3). Stipes central to slightly eccentric, 20–40 mm long, 10–20 mm wide, subcylindrical, slightly tapering downward, solid; surface dry, glabrous, white, unchanging when bruised; basal mycelium white. Odor indistinctive. Taste none.

Basidiospores 4–5.5 × 3.5–4.5 μm, av. 4.73 ± 0.38 × 3.88 ± 0.35, Q = 1.1–1.42, Q_m_ = 1.22 ± 0.09, broadly elliptical to subglobose, smooth, thin-walled, hyaline, some with granular contents. Basidia 30–40 × 4–7 μm, subclavate to clavate, 2- to 6-spored, sterigmata 3–5 μm long. Basidioles numerous, subcylindrical or subclavate, sometimes covered with a yellowish granular substance at the apex. Cystidia absent. Hyphae of spine apex cylindrical, thin-walled, yellowish to hyaline in KOH, 3–6 μm wide. Pileipellis a mixocutis, composed of cylindrical, thin-walled, branched hyphae, 2–5 μm wide. Stipitipellis composed of parallel, thin-walled hyphae, 2–6 μm wide. Clamp connections present.

Habitat and distribution: solitary or gregarious in broad-leaved or mixed forests mainly dominated by Fagaceae trees (such as *Castanopsis eyrei*, *C. fissa*, *C. lamontii*, and *Cyclobalanopis chungii*). Known from southern China and Japan.

Specimens examined: CHINA. Guangdong Province, Meizhou City, Fengshun Town, 15 February 2022, Wangqiu Deng and Ting Li (GDGM87013); Hubei Province, Yichang City, Changyang County, Langping Town, 28 October 2023, Qinfang Luo (GDGM93011).

Notes—*Hydnum cremeoalbum*, recently described from Japan [6], is firstly recorded in China in this study. The morphological description is short in Niskanen et al. [6]. Sugawara et al. [24] gave a detailed description of *H. cremeoalbum* after re-studying the type specimen and examining a large number of specimens collected from Japan. Morphologically, *H. cremeoalbum* is mainly characterized by the cream-white pileus which first changes to pale ocher when bruised and finally staining greenish to bluish green, the pale cream to pale orange spines, and the broadly ellipsoid basidiospores measuring 5–6.5 × 4–5.5 µm [24]. It is worth noting that the discoloration of basidiocarps from greenish to bluish green is rare in *Hydnum* species [24]. 

Phylogenetic analyses (Figure 1 and Figure 2) shows that two specimens from China are nested into the *H. cremeoalbum* clade with well-supported evidence. Morphological comparisons also showed that most features of our specimens are consistent with the description of *H. cremeoalbum*, except that the discoloration was not observed in the present study.

*Hydnum longipes* Ming Zhang & C.Q. Wang sp. nov.; Figure 3b and Figure 5.

Fungal Name: FN571736

Etymology—Refers to the species with a long stipe.

Type—CHINA. Yunnan Province, Shangrila, Bitahai National Natural Reserve, 27°49′ N, 99°57′ E, elev. 3560 m, on soil under *Quercus aquifolioides* Rehd. et Wils. and *Pinus densata* Mast. dominated forests, 2 September 2020, Ming Zhang (GDGM82458). 

Diagnosis—*Hydnum longipes* can be distinguished from the other *Hydnum* species by the combination of orange-to-reddish orange pileus, orange-white spines, a longer stipe, and subglobose basidiospores measuring 8–10 × 7–8.5 μm.

Basidiocarps small sized, solitary to scattered. Pileus 20–30 mm broad, convex to subapplanate, usually depressed in the center, margin thin, even, decurved at first; surface dry, glabrous, irregularly bumpy or mottled, orange, light orange, pastel red to reddish-orange (5A7–8A7, 5A5–8A5), changing to orange-white when bruised. Context 1–2 mm thick, orange-white (5A2). Hymenophore spinose, adnate, not decurrent; spines conical to aciculate, crowded, subulate, orange-white to pale orange (4A2–5A2, 4A3–5A3), 1–3 mm long, shorter near the pileus margin. Stipe central to slightly eccentric, 50–80 mm long, 3–6 mm wide, subcylindrical, slightly enlarged downward, hollow; surface dry, glabrous, orange-white to pale orange (4A2–5A2, 4A3–5A3), slightly changing to orange-white when bruised; basal mycelium white. Odor indistinctive. Taste none. 

Basidiospores (7.5)8–10(10.5) × (6) 7–8.5(9) μm, av. 9.32 ± 0.77 × 7.63 ± 0.87, Q = 1.05–1.43, Q_m_ = 1.23 ± 0.11, broadly elliptical to subglobose, smooth, thin-walled, hyaline, some with granular contents. Basidia 45–55 × 7–12 μm, subclavate to clavate, 2- to 6-spored, sterigmata 4–8 μm long, 1–2 μm wide at base, slightly curving. Basidioles numerous, subcylindrical or subclavate, usually covered with a yellowish granular substance at the apex. Cystidia absent. Hyphae of spine apex are cylindrical, thin-walled, yellowish to hyaline in KOH, 3–6 μm wide. Pileipellis a cutis composed of cylindrical, thin-walled, subparallel, rarely branched hyphae, 5–16 μm wide; terminal cells 70–135 × 5–16 μm. Stipitipellis composed of parallel, thin-walled hyphae, 3–8 μm wide. Clamp connections present.

Habitat and distribution—Growing solitarily or gregariously in subalpine mixed forests mainly dominated by *Quercus aquifolioides* and *Pinus densata*. So far known from southwestern China. 

Notes—Phylogentic analysis showed that *H. longipes* formed a distinct lineage in subg. *Rufescentia*, and is closely related to *H. canadense*, *H. mulsicolor*, and *H. submulsicolor*. Based on the morphological characteristics given above and the phylogentic results, *H. longipes* should be a member within subsect. *Mulsicoloria*.

Morphologically, *H. canadense* differs from *H. longipes* by its cream-colored spines, shorter and white-to-cream stipe, globose-to-subglobose basidiospores (7–9 × 7–9 μm, Q = 1–1.11), smaller basidia; in addition, *H. canadense* is only known in high-elevation conifer forests in North America [25]. *Hydnum mulsicolor*, originally reported from Slovenia, differs in having larger basidiocarps (pileus 30–45 mm broad), bright orange-to-tan pileus usually distinctly umbilicate at center, strongly decurrent pinkish-cream spines, a shorter stipe with distinct rhizomorphs at the base, and smaller basidiospores (6.5–8.5 × 6–8.5 μm) [6,25]. *Hydnum submulsicolor*, originally described form Canada, differs by its relatively larger basidiocarps (pileus 30–50 mm broad), orange ochraceous to more brownish ochraceous pileus, and the whitish-to-pale ochraceous brown spines [6].

*Hydnum melitosarx* Ruots., Huhtinen, Olariaga, Niskanen, Liimat. & Ammirati; Figure 3c and Figure 6.

Basidiocarps small sized, solitary, scattered to gregarious. Pileus 20–40 mm broad, convex to subapplanate, margin thin, even, decurved at first; surface dry, glabrous, irregularly bumpy or mottled, reddish-yellow, light yellow, light orange to orange, pale yellow to pale orange (4A5–5A5, 4A6–5A6). Context 2–4 mm thick, white, unchanging or slightly changing pale orange when exposed. Hymenophore spinose, adnate; spines conical to aciculate, crowded, subulate, white, yellowish white to pale orange (3A2–5A2, 3A3–5A3), 2–5 mm long, shortest near the pileus margin. Stipes central to slightly eccentric, 20–40 mm long, 4–8 mm wide, subcylindrical, solid; surface dry, glabrous, white, to orange-white (5A2), unchanging or changing pale orange when handled; basal mycelium white. Odor indistinctive. Taste none. 

Basidiospores 8–11 × 7–10 μm, av. 9.37 ± 0.65 × 8.71 ± 0.30, Q = 1.0–1.28, Q_m_ = 1.07 ± 0.06, broadly ellipsoid to ovia, smooth, thin-walled, hyaline, some with granular contents. Basidia 40–70 × 10–15 μm, subclavate to clavate, 2- to 6-spored, sterigmata 3–6 μm long, 1–1.5 μm wide at base. Basidioles numerous, subcylindrical or subclavate. Cystidia absent. Hyphae of spine apex cylindrical, thin-walled, yellowish to hyaline in KOH, 2–7 μm wide. Pileipellis composed of cylindrical hyphae, thin-walled, subparallel, rarely branched, 2–13 μm wide. Stipitipellis composed of parallel hyphae, thin-walled, 3–12 μm wide. Clamp connections present. 

Habitat and distribution—Growing in solitary or gregarious conditions under broad-leaved forests mainly dominated by Fagaceae trees (such as *Quercus aquifolioides* and *Q. pannosa*) or in mixed forests dominated by *Picea* trees mixed with a small number of *Betula*, *Alnus*, *Salix,* and *Populus*. Known from Asia, Europe, and North America.

Specimens examined—CHINA. Sichuan Province, Aba Tibetan and Qiang Autonomous Prefecture, Jiuzhaigou County, 20 September 2020, elev. 2600 m, Ming Zhang (GDGM84518); Sichuan Province, Aba Tibetan and Qiang Autonomous Prefecture, Lixian County, Miaro Scenic Area, 4 August 2020, elev. 2600 m, Chaoqun Wang (GDGM81873, GDGM81826).

Notes—As Niskanen et al. [6] mentioned, the Chinese *H. melitosarx* is also characterized by its pale orange-brown ochraceous to orange-brown ochraceous pileus, whitish spines, relatively longer stipe. However, the Chinese specimens have larger basidiospores (8–11 × 7–10 μm and av. = 9.37 × 8.71 μm vs. 7.0–8.6 × 6.4–7.8 μm and av. = 7.9 × 7.2 μm) [6].

In the present study (Figure 1 and Figure 2), two specimens (GDGM81873 and GDGM84518), together with several sequences labelled as “*Hydnum rufescens*” from China, are well nested into the *H. melitosarx* clade. Although the distribution of this species in China was found by previous analysis [6], no detailed morphology studies have been conducted. This study is the first to describe this species in detail based on Chinese materials. It is worth mentioning that *H. melitosarx* is a pan-Arctic distributed species, so far known from Andorra, China, Estonia, Finland, Germany, Solenia, Sweden, and the USA [6]. 

*Hydnum microcarpum* Ming Zhang sp. nov. Figure 3f and Figure 7.

Fungal Name: FN571762 

Etymology—‘*microcarpum*’ refers to the species with a smaller basidiocarp.

Type—CHINA. Guangdong Province, Huizhou City, Luofushan Scenic Spot, 23°15′ N, 114°00′ E, elev. 600 m, on soil under mixed forests dominated by Fagaceae trees (such as *Castanopsis cuspidata*, *C. eyrei*, *C. fabri*, *C. fissa,* and *Lithocarpus* spp.) mixed with *Pinus massoniana* Lamb., 1 March 2022, Guorui Zhong (GDGM87902). 

Diagnosis—This species is characterized by its small basidiocarps, yellowish white to orange-white pileus changing from brownish orange to brownish red when bruised, yellowish white spines, and relatively small basidiospores.

Basidiocarps small sized, solitary to scattered. Pileus 10–20 mm broad, convex to subapplanate, margin thin, even, decurved at first; surface dry, glabrous, white, yellowish white to orange-white (3A2–5A2), changing from brownish orange to brownish red when bruised or dry (5C5–8C5). Context 1–2 mm thick, yellowish white (4A2). Hymenophore spinose, adnate to subdecurrent, spines crowded, subulate, acute, straight, solitary, evenly distributed, white to yellowish white (3A2–5A2) when fresh, grayish yellow to grayish orange (4B5–6B5) when dry; 0.5–1.5 mm long, shorter toward pileus margin. Stipes central to slightly eccentric, 15–25 mm long, 4–6 mm wide, subcylindrical, surface dry, glabrous or glandular, white, staining orange-white (4A2–5A2) when bruised; basal mycelium white, usually forming rhizomorphs at base. Odor indistinctive. Taste unrecorded.

Basidiospores 5–6(–6.5) × 5–5.5(–6) μm, av. 5.78 ± 0.31 × 5.26 ± 0.36, Q = 1–1.2, Q_m_ = 1.10 ± 0.08, broadly elliptical to subglobose, smooth, thin-walled, hyaline, some with granular contents. Basidia 27–40 × 7–10 μm, subclavate to clavate, 4- or 6-spored, sterigmata 4–6 μm long, 1 μm wide at base. Basidioles numerous, subcylindrical or subclavate, usually covered with yellowish granular substance at apex. Cystidia absent. Hyphae of spine apex cylindrical, thin-walled, yellowish to hyaline in KOH, 2–6 μm wide. Pileipellis composed of cylindrical hyphae, thin-walled, subparallel, rarely branched; terminal elements rounded at apex, cells 80–135 × 7–12 μm. Stipitipellis composed of parallel hyphae, thin-walled, 2–7 μm wide. Clamp connections present.

Habitat and distribution—Growing in solitary or gregarious conditions under mixed forests in the subtropical region of southern China, which is mainly dominated by *Castanopsis* trees and *Lithocarpus* trees mixed with a small number of *Pinus massoniana*. So far only known from southern China. 

Notes—In both the ITS and multilocus phylogenetic analyses (Figure 1 and Figure 2), the new species *H. microcarpum* formed an independent lineage, close to *H. alboluteum* R. Sugaw. & N. Endo, *H. brevispinum* T. Cao & H.S. Yuan, and *H. tenuistipitum* T. Cao & H.S. Yuan. Based on the morphological features given above and the phylogenetic evidence, the new species *H. microcarpum* is a member of subg. *Brevispina*, which has been recently established in Cao et al. [2].

Morphologically, *H. alboluteum* differs by its larger basidiocarps (pileus 20–55 mm broad), longer spines up to 6 mm long, and larger basidiospores (6.5–8 × 5.5–7.5 µm) [24]. *Hydnum brevispinum* differs by its more broadly elliptical basidiospores (5–5.8 × 3.8–4.8 μm) with large Q values (Q = 1.27–1.31) [2], and the difference in ITS sequence, with a similarity ratio of only 94.35% with *H. microcarpum* in the Blast search. *Hydnum tenuistipitum* can be easily distinguished by its larger basidiospores (6.8–7.2 × 5.5–6.5 μm) and basidia (45–63 × 3–12 μm) [2]. In addition, *H. minum* Yanaga & N. Maek., recently reported from Japan, also resemble *H. microcarpum* with small basidiocarps, but differ in having whitish to cream-yellow spines, and smaller basidiospores (4.5–5.5 × 3–4.5 mm) with a larger Q value (av. Q = 1.3) [26].

*Hydnum orientalbidum* R. Sugaw. & N. Endo; Figure 3d and Figure 8.

Basidiocarps small to medium sized, solitary, scattered to gregarious. Pileus 30–60 mm broad, convex to subapplanate, margin thin, even, decurved at first; surface dry, glabrous, white. Context 3–4 mm thick, white, unchanging when exposed. Hymenophore spinose, adnate; spines conical to aciculate, crowded, subulate, white, 2–3 mm long, shortest near the pileus margin. Stipes central to slightly eccentric, 30–50 mm long, 8–15 mm wide, subcylindrical, solid; surface dry, glabrous, white; basal mycelium white. Odor indistinctive. Taste none.

Basidiospores 4–5 × 3.5–4.5 μm, av. 4.65 ± 0.41 × 3.96 ± 0.37, Q = 1.1–1.28, Q_m_ = 1.17 ± 0.06, broadly ellipsoid to ovia, smooth, thin-walled, hyaline, some with granular contents. Basidia 20–40 × 5–7 μm, subclavate to clavate, 2- to 6-spored, sterigmata 3–6 μm long, 1–1.5 μm wide at base. Basidioles numerous, subcylindrical or subclavate. Cystidia absent. Hyphae of spine apex cylindrical, thin-walled, yellowish to hyaline in KOH, 2–6 μm wide. Pileipellis composed of cylindrical hyphae, thin-walled, subparallel, rarely branched, 5–13 μm wide. Stipitipellis composed of parallel hyphae, thin-walled, 2–5 μm wide, containing brownish cytoplasmic pigment. Clamp connections present.

Habitat and distribution—Growing in solitary or gregarious conditions under broad-leaved forests mainly dominated by *Castanopsis* and *Quercus* trees. Known from China and Japan.

Specimen examined—CHINA. Sichuan Province, Bazhong City, Guangwushan National Geopark, 21 October 2023, elev. 2500 m, Yi Yang (GDGM93019). Chongqing City, Wushan Town, 14 November 2023, Jiaxi Pan (GDGM93480); Zhejiang Provice, Hangzhou City, Hangzhou Botanical Garden, 6 April 2023, elve. 80 m, Qingqing Huang (GDGM91301).

Notes—The Chinese specimens share consistent characteristics with previous descriptions of *H. orientalbidum* [24], but the color change is not observed in the present study. Phylogenetic analyses on the basis of ITS and multilocus datasets show that three specimens from China are well nested into the *H. orientalbidum* clade, which is located at the basal branch. 

*Hydnum pinicola* R. Sugaw. & N. Endo; Figure 3e and Figure 9.

Basidiocarps small to medium sized, solitary, scattered to gregarious. Pileus 25–60 mm broad, convex to subapplanate, margin thin, even, decurved at first; surface dry, glabrous, irregularly bumpy or mottled, white, yellowish white, pale yellow to pale orange (4A2, 4A3–5A3). Context 5–10 mm thick, white, unchanging when exposed. Hymenophore spinose, adnate; spines conical to aciculate, crowded, subulate, orange-white pale orange (5A2–5A3) when fresh, pinkish white to pale red (7A2–8A2) when dry or old; 1.5–3 mm long, shortest near the pileus margin. Stipes central to slightly eccentric, 20–30 mm long, 8–25 mm wide, subcylindrical, solid; surface dry, glabrous, white, unchanging when handled; basal mycelium white. Odor indistinctive. Taste bitter.

Basidiospores 4–6 × 3.5–5 μm, av. 4.72 ± 0.53 × 4.02 ± 0.31, Q = 1.0–1.25, Q_m_ = 1.17 ± 0.08, broadly ellipsoid to ovia, smooth, thin-walled, hyaline, some with granular contents. Basidia 23–43 × 6–8 μm, subclavate to clavate, four- to eight-spored, sterigmata 2–6 μm long, 1–1.5 μm wide at base. Basidioles numerous, subcylindrical or subclavate, sometimes covered with a yellowish granular substance at the apex. Cystidia absent. Hyphae of spine apex cylindrical, thin-walled, yellowish to hyaline in KOH, 3–7 μm wide. Pileipellis composed of cylindrical hyphae, thin-walled, subparallel, rarely branched, 2–17 μm wide. Stipitipellis composed of parallel hyphae, thin-walled, 2–12 μm wide. Clamp connections present.

Habitat and distribution—Growing in solitary or gregarious conditions under mixed forests mainly dominated by Fagaceae trees and mixed with *Pinus* trees. Currently known from China and Japan. 

Specimen examined—CHINA. Hubei Province, Huanggang City, Luotian County, Shengli Town, 22 October 2023, elev. 800, Xiaomei Peng (GDGM93020). Yunnan Province, Lijiang City, Yulong County, Jiuhe Village, 1 September 2020, elev. 2400 m, Ming Zhang (GDGM83047).

Notes—*Hydnum pinicola*, originally described from Japan, is characterized by its whitish to cream basidiocarps, pale yellow or pale orange spines, and relatively smaller basidiospores (4.5–5.5 × 4–5 µm) [24]. Our specimens generally matched the original description of *H. pinicola* in morphology and existed in similar habitats. In both the ITS and multilocus phylogenetic analyses (Figure 1 and Figure 2), the new Chinese samples fell into the *H. pinicola* clade with strong support (BS = 100%). In short, the specimens from southern China are identified as *H. pinicola*, which was first recorded in China.

*Hydnum sinorepandum* Ming Zhang & C.Q. Wang sp. nov. Figure 3g,h and Figure 10.

Fungal Name: FN571763

Etymology: Refers to the species reported from China, similar to *H. repandum*.

Type—CHINA. Yunnan Province, Shangrila, Bitahai National Natural Reserve, 27°49′ N, 99°57′ E, elev. 3560 m, on soil under broad-leaved and mixed forests, 2 September 2020, Ming Zhang (GDGM82445). 

Diagnosis—This species is characterized by its relatively large basidiocarps, yellowish white to light orange pileus, bitter context changing to orange-white when bruised, yellowish white spines up to 7 mm long, robust stipe, broadly ellipsoid basidiospores, and sub-alpine distribution.

Basidiocarps medium to large sized, solitary to scattered. Pileus 40–120 mm broad, convex at first, subapplanate when mature, usually depressed in the center or of an irregular shape, margin thin, decurved at first; surface dry, glabrous, irregularly bumpy or mottled, yellowish white, pale yellow, light yellow to light orange (4A1–4A2, 4A4–6A4). Context 3–8 mm thick, white, unchanging or slightly changing from orange-white to pale orange (5A2–5A3) when exposed. Hymenophore spinose, adnate; spines conical to aciculate, crowded, subulate, white, yellowish white, pale yellow to pale orange (4A2, 4A3–5A3), 3–7 mm long, shorter toward the pileus margin. Stipes central to slightly eccentric, 50–100 mm long, 7–20 mm wide, subcylindrical, solid; surface dry, glabrous, white to yellowish white (4A2); basal mycelium white. Odor indistinctive. Taste mild or a little bitter.

Basidiospores (7.5)8–9(10) × 6.5–7.5 μm, av. 8.16 ± 0.53 × 7.12 ± 0.42, Q = 1.0–1.38, Q_m_ = 1.15 ± 0.09, broadly ellipsoid, smooth, thin-walled, hyaline, some with granular contents. Basidia 40–60 × 8–15 μm, subclavate to clavate, 4- to 8-spored, sterigmata 2–7 μm long, 1–3 μm wide at base, slightly curving. Basidioles numerous, subcylindrical or subclavate, usually covered with a yellowish granular substance at the apex. Cystidia absent. Hyphae of spine apex cylindrical, thin-walled, yellowish to hyaline in KOH, 4–7 μm wide. Pileipellis composed of cylindrical hyphae, thin-walled, subparallel, rarely branched; terminal cells up to 150 μm long, 4–12 μm wide. Stipitipellis composed of subparallel to slightly interwoven hyphae, thin-walled, 4–10 μm wide; terminal cell cylindrical, subclavate or irregular shape. Clamp connections present.

Habitat and distribution—Growing in solitary or gregarious conditions under broad-leaved forests of *Quercus aquifolioides*, *Betula albosinensis*, *Populus tremula,* and *Rhododendron* spp.; and in mixed forests of *Quercus, Betula*, *Rhododendron* and *Abies georgei* in subalpine regions. Currently only known from southwestern China. 

Additional specimens examined—CHINA. Yunnan Province, Shangrila, Bitahai National Natural Reserve, 27°49′ N, 99°57′ E, elev. 3560 m, 2 September 2020, Ming Zhang (GDGM82416, GDGM82382); Same location, 4 August 2022, Ming Zhang (GDGM90134). 

Notes—Morphologically, *H. sinorepandum* can be easily misidentified as *H. repandum*; however, the latter differs by decurrent spines, smaller basidiospores (7.0–8.5 × 6.2–7.5 μm), smaller basidia (35–45 × 6–8 μm), and so far is only known to exist in Europe according to Niskanen et al. [6]. *Hydnum boreorepandum* and *H. repando-orientale* also resemble *H. sinorepandum.* However, *H. boreorepandum*, reported from Europe, differs by its white to pale cream pileus, tinged pale ochraceous brown to pale brownish orange when bruised, white to cream spines, a slender stipe with rhizomorphs at the base, and relatively smaller basidiospores (7.0–8.5 × 6.2–7.5 μm) [6,24]. *Hydnum repando-orientale*, reported from Japan, differs by its whitish to cream-white pileus and spines, and smaller basidiospores (7–9 × 5.5–8 μm) [6,24,27]. 

Both the ITS and multilocus phylogenetic analyses showed that the four Chinese specimens of *H. sinorepandum* in this study clustered together with several sequences labelled as “*Hydnum* sp. 8”, and that they formed a distinct lineage close to *H. sphaericum* within the subg. *Hydnum*. *Hydnum sphaericum*, also originally described from China, can be easily distinguished from *H. sinorepandum* by its smaller basidiocarps (pileus 20–35 mm broad in diam.) with subglobose pileus when young, shorter spines, and the distribution in subtropical regions of central China [2].

## 4. Discussion

In this study, seven *Hydnum* species, including three new species (i.e., *H. longipes*, *H. microcarpum*, and *H. sionrepandum*), were confirmed and described on the basis of morphological and molecular analyses in China. Up to now, 25 *Hydnum* species have been recorded in China, including 15 species originally reported from China [2]. A morphological comparison of *Hydnum* species in China is given in Table 1, including the main morphological characteristics and ecological information. 

Compared with previous morphological results, the number of *Hydnum* species now recorded in China has increased dramatically. This is mainly due to the overlapping morphological characteristics between *Hydnum* species (such as the size, shape, and color of the cap, tooth, and stipe); thus, recognition of *Hydnum* species based solely upon morphology is very difficult or impossible. 

In some species, it is noticeable that some morphology characteristics have intraspecific variations after observing and studying more specimens. For instance, the spore size of *H. melitosarx* from the northern and subtropical alpine areas of China is larger than that from southern China. The reason for this may be that there is more pressure to survive in the southern regions, and the fungi need to mature as quickly as possible to release more spores to maintain the population. In addition, discoloration (context changing from pale yellowish green to bluish green when bruised) was observed in some Japanese *H. cremeoalbum* samples [24] but not observed in Chinese specimens in the present study. Many *Hydnum* species will undergo color changes after injury or handling, but this is not a stable characteristic and it is not a wise choice to distinguish *Hydnum* species based on color change.

Recently, molecular sequences appeared to be very effective in distinguishing species of this genus, and some cryptic species have been discovered. For example, “*Hydnum repandum*” had been widely reported in China [8,9,10,30]; however, recent phylogenetic studies proved that the Chinese “*H. repandum*” is a species complex, containing at least four species, such as, *H. berkeleyanum*, *H. melitosarx*, *H. orientalbidum,* and *H. sinorepandum*. In addition, *H. repandum* is so far only known from Europe.

In the multilocus phylogenetic tree (Figure 2), the genus *Hydnum* is an independent monophyletic group, in which four subgenera (subg. *Pallidi*, subg. *Hydnum*, subg. *Parvispina*, and subg. *Rufescentes*) are highly supported, and the subg. *Alba* is not monophyletic, which is similar to the results of previous studies [2,23,24]. However, no stable morphological characteristics have been found to distinguish each subgenus. For example, traditionally, subg. *Alba* is characterized by whitish to cream-colored pileus, a whitish stipe which stains yellowish when handled, and the features also seen in some species of subg. *Hydnum,* subg. *Pallidi* and subg. *Rufescentes.* In addition, the presence of relatively small basidiospores (usually less than 7 mm long) is another characteristic to distinguish subg. *Alba* from others, while these characteristics are also present in subg. *Parvispina*.

Most species of *Hydnum* have restricted distribution areas according to previous studies [5,24,27] and this study. For example, *H. subalpinum* and *H. sinorepandum* are only distributed in sub-alpine regions of Asia; *H. repandum* is only known in Europe [6]; and *H. alboaurantiacum*, *H. cuspidatum*, *H. ferruginescens,* and *H. subtilior* appeared endemic to North America [25]. Currently, only *H. melitosarx* has been confirmed with a holarctic distribution, so far known from Andorra, China, Estonia, Finland, Germany, Solenia, Sweden, and the USA. It is worth mentioning that *Hydnum* species can be found in temperate to subtropical regions in China, but it is very rare in tropical regions. 

In recent years, a large number of *Hydnum* species have been found in Asia, making Asia the continent with the highest species diversity of *Hydnum* species [2,23,24,26,27], and most known *Hydnum* species are endemic to Asia. China, as one of the mega-biodiverse countries, has numerous plants that can symbiosis with *Hydnum*, and it can be predicted that with the deepening of investigations and research, many more *Hydnum* species will be discovered and reported from this country.

Key to species of *Hydnum* in China
1. Pileus white, cream white, or pale yellow21.Pileus orange, reddish orange, or reddish brown112.Basidiocarps medium to large; pileus > 60 mm broad32.Basidiocarps small; pileus ≤ 60 mm broad43.Distributed in sub-alpine regions*H. sinorepandum*3.Distributed in subtropical regions, elevation < 1000 m*H. cremeoalbum*4.Pileus ≤ 25 mm broad54.Pileus usually > 25 mm broad75.Basidiospores > 5 μm long on average65.Basidiospores < 5 μm long on average *H. minum*6.Spines 0.5–1.5 mm long, white to yellowish white when fresh, grayish yellow to grayish orange when dry*H. microcarpum*6.Spines 0.2–0.8 mm long, pure white when fresh, yellowish white when dry*H. brevispinum*7.Basidiospores > 7.5 μm long on average87.Basidiospores < 7.5 μm long on average98.Distributed in northeastern China; basidiospores av. 9.07 × 7.04 μm*H. flabellatum*8.Distributed in southwestern China; basidiospores av. 7.75 × 6.01 μm*H. flavidocanum*9.Spines white; basidiospores av. 4.65 × 3.96 μm*H. orientalbidum*9.Spines orange-white when fresh1010.In mixed forests; basidiospores av. 4.72 × 4.02 μm*H. pinicola*10.In broad-leaved forests; basidiospores av. 7.08 × 6.09 μm*H. tenuistipitum*11.Basidiocarps small; pileus < 20 mm broad, orange-white to greyish orange; basidiospores av. 9.81 × 9.03 μm*H. longibasidium*11.Basidiocarps medium to large; pileus ≥ 20 mm broad1212.Basidiospores > 9 μm long on average1312.Basidiospores < 9 μm long on average1513.Holarctic distribution; pileus reddish yellow, light orange to orange*H. melitosarx*13.Only distributed in Asia1414.Distributed in subalpine regions of southwestern China; pileus orange to reddish orange, changing orange-white when bruised; spines adnate, orange-white to pale orange*H. longipes*14.Distributed in coniferous forests of northwestern China; pileus orange-white to pale orange; spines light yellow when fresh*H. pallidocroceum*15.Pileus can up to 80 mm broad, pale orange to light orange; spines up to 9 mm long; distributed in temperate regions of Asia*H. berkeleyanum*15.Pileus less than 80 mm broad1616.Spines decurrent1716.Spines non-decurrent to subdecurrent1817.Pileus very pale to medium orange ocher; basidiospores av. 7.5 × 7 μm; distributed in coniferous forests*H. jussii*17.Pileus orange-white to pale orange; basidiospores av. 8.75 × 6.99 μm; distributed in broad-leaved forests*H. pallidomarginatum*18.Distributed in broad-leaved forests in central China1918.Distributed in mixed forests2019.Spines white, 0.5–3 mm long; pileus orange-white*H. sphaericum*19.Spines orange-white, 2–6 mm long; pileus light orange to brownish orange*H. tangerinum*20.Pileus orange to brown; spines orange-white when fresh, up to 5 mm long; basidiospores av. 8.64 × 8.17 μm with Q value = 1.05–1.09*H. ventricosum*20.Pileus ocher to light ocher; spines pale ocher, 1–1.7 mm long; basidiospores av. 8.2–8.7 × 6.4–6.8 μm with Q value = 1.27–1.3*H. vesterholtii*

## Figures and Tables

**Figure 1 jof-10-00098-f001:**
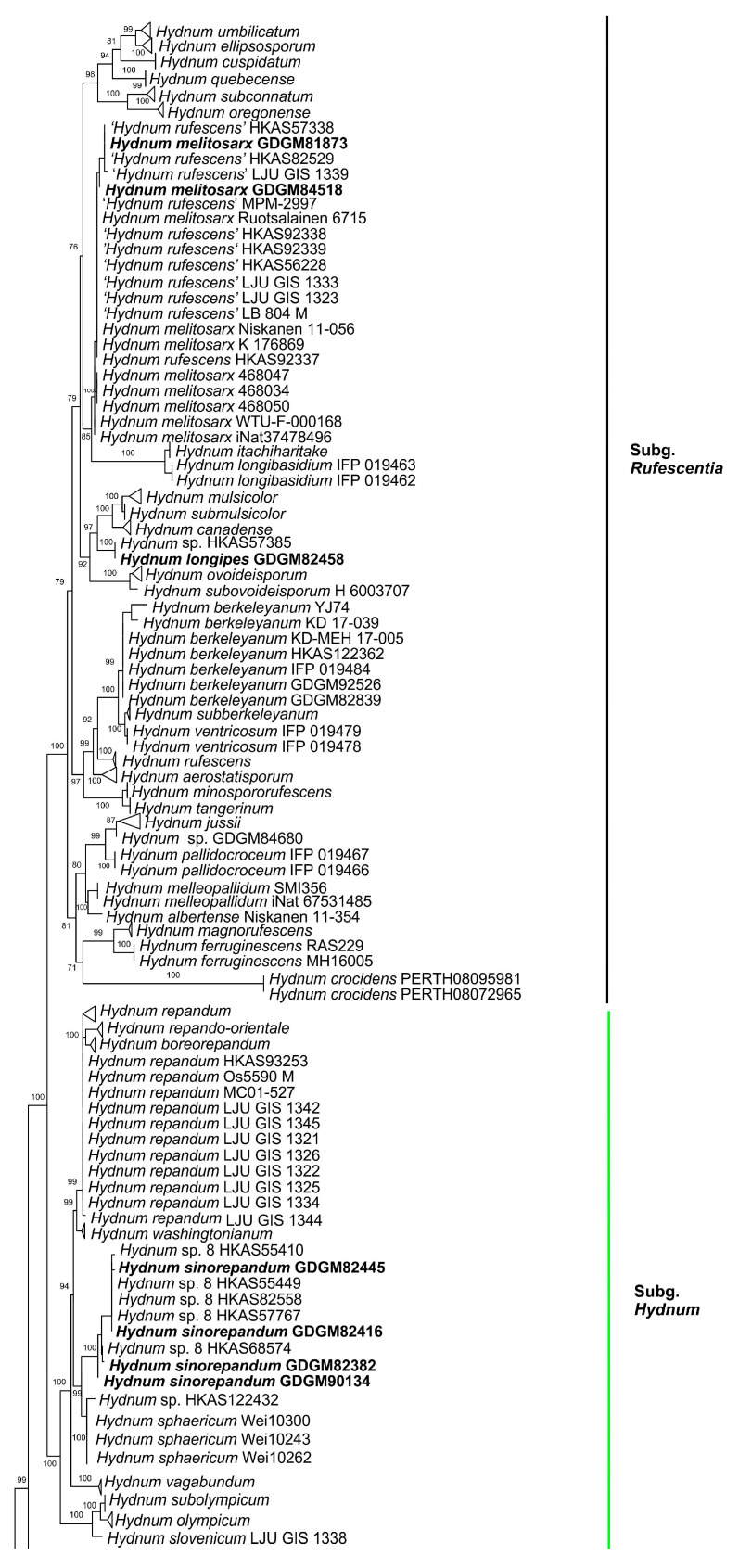
Maximum likelihood phylogenetic tree of *Hydnum* inferred from the ITS dataset. Bootstrap values (>70%) are shown above or below supported branches. Specimens from the present study are indicated in bold.

**Figure 2 jof-10-00098-f002:**
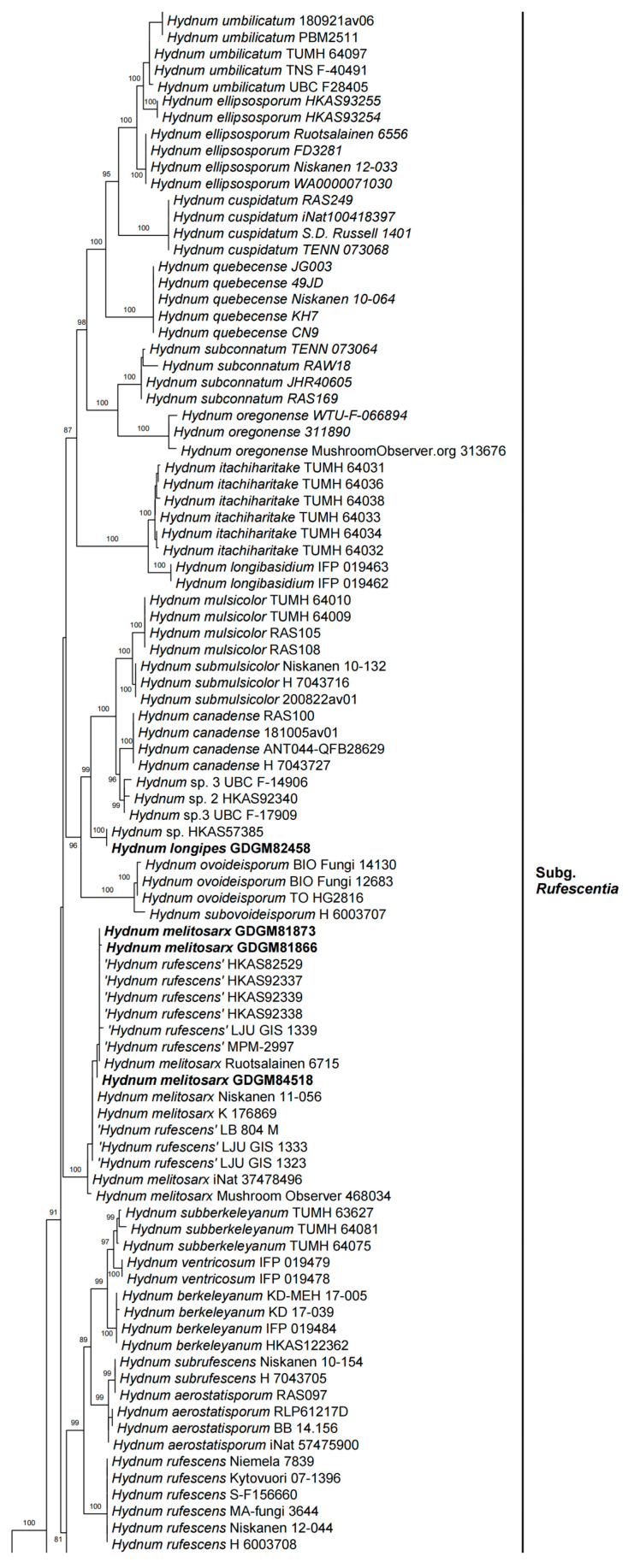
Maximum likelihood phylogenetic tree of *Hydnum* inferred from the ITS-LSU-*tef*1 dataset. Bootstrap values (>70%) are shown above or below supported branches. Specimens from this study are indicated in bold.

**Figure 3 jof-10-00098-f003:**
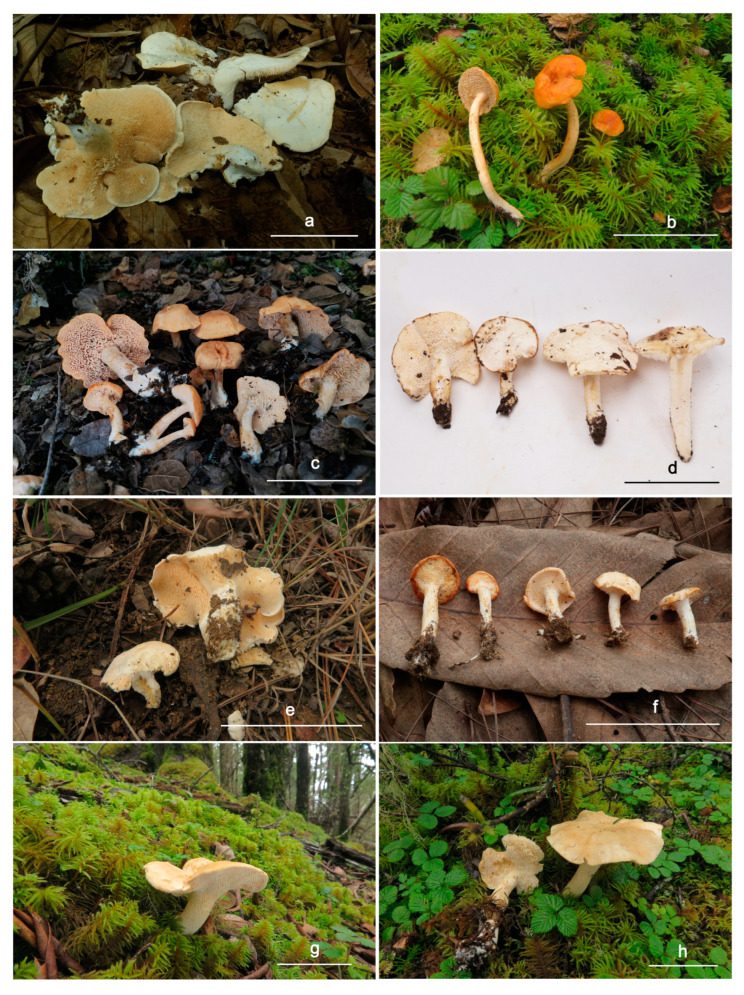
Fresh basidiocarps of *Hydnum* species. (**a**) *H. cremeoalbum* (GDGM87013). (**b**) *H. longipes* (GDGM82458, Type). (**c**) *H. melitosarxm* (GDGM81866). (**d**) *H. orientalbidum* (GDGM93019). (**e**) *H. pinicola* (GDGM83047). (**f**) *H. microcarpum* (GDGM87902, Type). (**g**,**h**) *H. sinorepandum* (**g**). GDGM82445, Type; (**h**). GDGM82382). Bars: (**a**–**h**) = 50 mm.

**Figure 4 jof-10-00098-f004:**
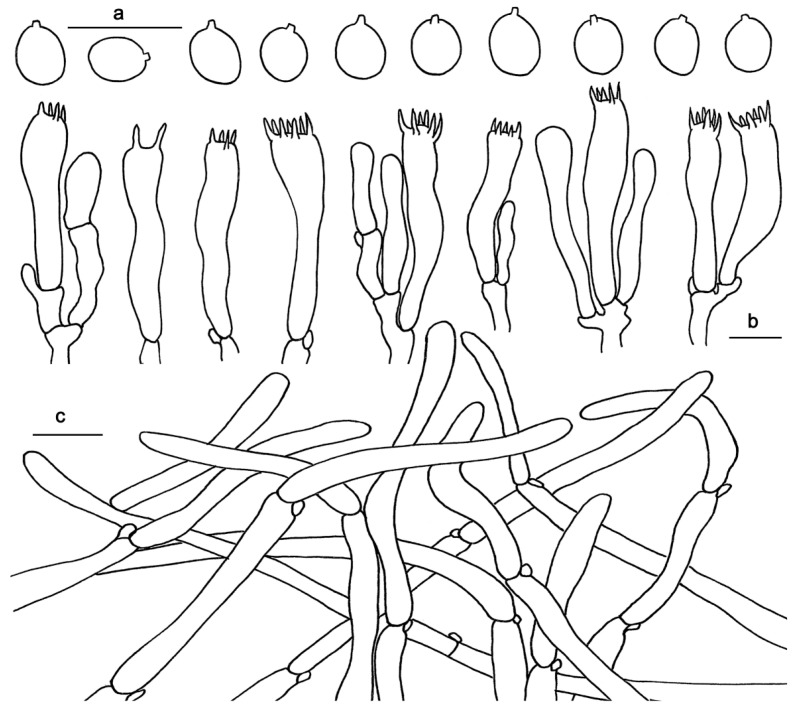
Microscopic features of *Hydnum cremeoalbum* (GDGM87013). (**a**) Basidiospores. (**b**) Basidia and basidioles. (**c**) Pileipellis terminal hyphae with clamp connections. Bars: (**a**–**c**) = 10 μm.

**Figure 5 jof-10-00098-f005:**
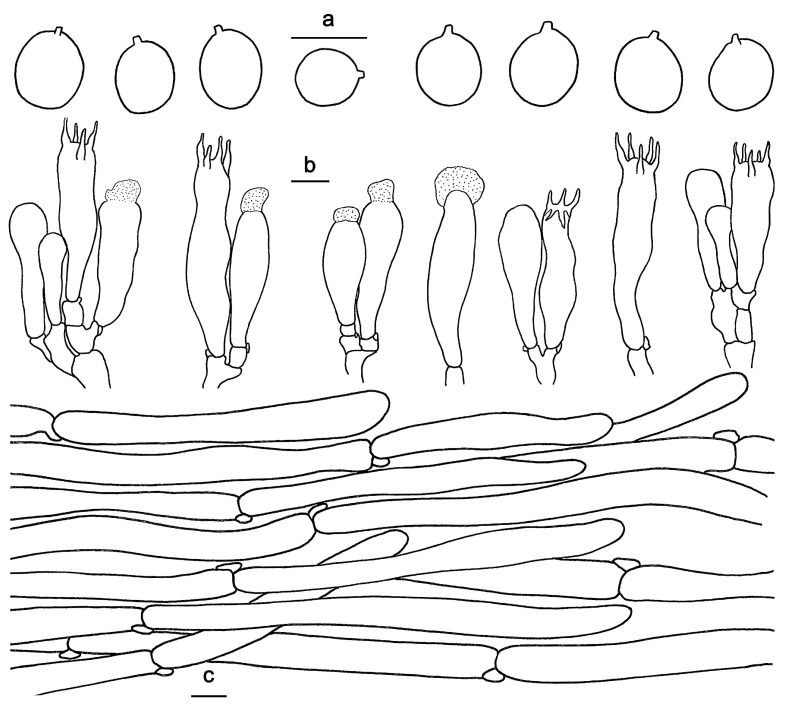
Microscopic features of *Hydnum longipes* (GDGM82458, Holotype). (**a**) Basidiospores. (**b**) Basidia and basidioles usually covered with a yellowish granular substance. (**c**) Pileipellis terminal hyphae with clamp connections. Bars: (**a**–**c**) = 10 μm.

**Figure 6 jof-10-00098-f006:**
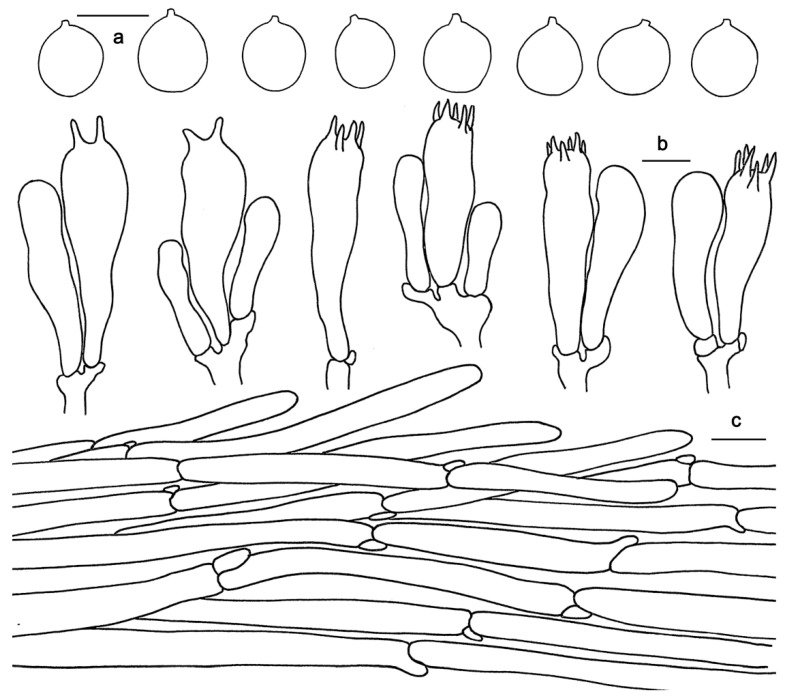
Microscopic features of *Hydnum melitosarx* (GDGM84518). (**a**) Basidiospores. (**b**) Basidia and basidioles. (**c**) Pileipellis terminal hyphae with clamp connections. Bars: (**a**–**c**) = 10 μm.

**Figure 7 jof-10-00098-f007:**
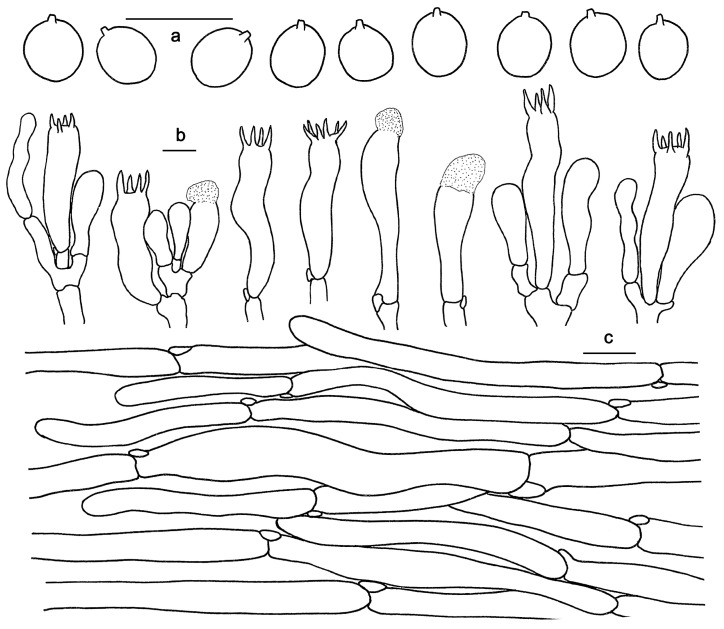
Microscopic features of *Hydnum microcarpum* (GDGM87902). (**a**) Basidiospores. (**b**) Basidia and basidioles usually covered with a yellowish granular substance. (**c**) Pileipellis terminal hyphae with clamp connections. Bars: (**a**–**c**) = 10 μm.

**Figure 8 jof-10-00098-f008:**
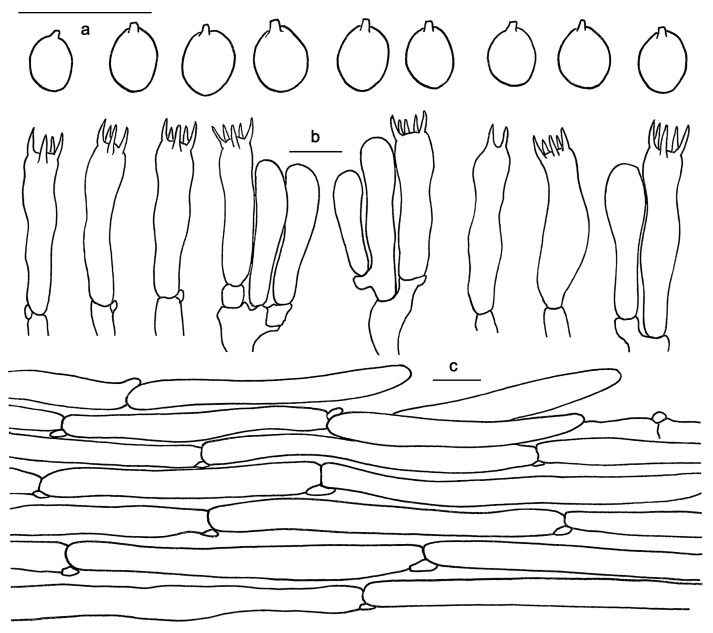
Microscopic features of *Hydnum orientalbidum* (GDGM93019). (**a**) Basidiospores. (**b**) Basidia and basidioles. (**c**) Pileipellis terminal hyphae with clamp connections. Bars: (**a**–**c**) = 10 μm.

**Figure 9 jof-10-00098-f009:**
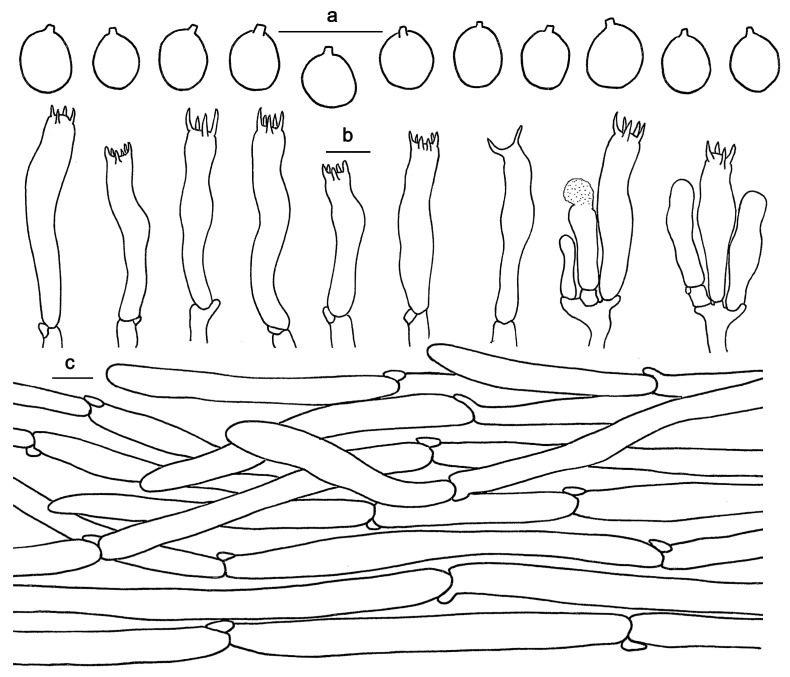
Microscopic features of *Hydnum pinicola* (GDGM93020). (**a**) Basidiospores. (**b**) Basidia and basidioles sometimes covered with a yellowish granular substance. (**c**) Pileipellis terminal hyphae with clamp connections. Bars: (**a**–**c**) = 10 μm.

**Figure 10 jof-10-00098-f010:**
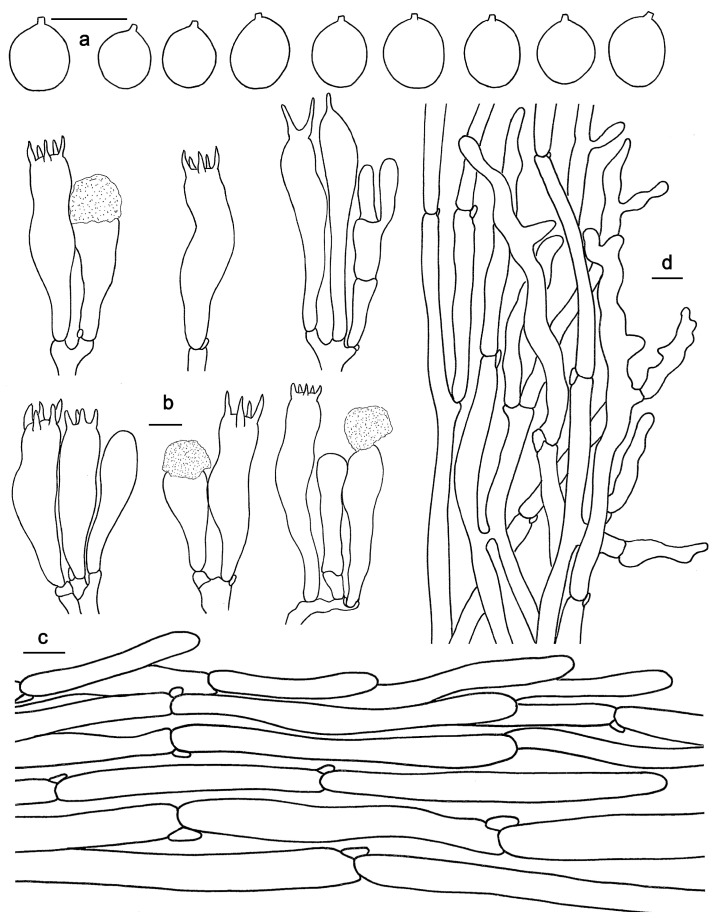
Microscopic features of *Hydnum sinorepandum* (GDGM82445). (**a**) Basidiospores. (**b**) Basidia and basidioles usually covered with a yellowish granular substance. (**c**) Pileipellis terminal hyphae with clamp connections. (**d**) Stipitipellis. Bars: (**a**–**d**) = 10 μm.

**Table 1 jof-10-00098-t001:** A morphological comparison of *Hydnum* species in China.

Species	Pileus	Spines	Spores Size	Habit and Distribution	References
*H. berkeleyanum*	25–80 mm wide; pale orange to light orange with a brown center, pale orange to light orange toward margin.	Spines adnexed to subdecurrent; initially white (A1), becoming orange-white to pale orange at maturity; 2–9 mm long.	7.5–8.4–8.5–9.5 × 6.9–7.9–8–8.8 µm; Q = 1.01–1.17	In temperate mixed forest; Known from India and China	[4,28]
*H. brevispinum*	10–15 mm wide; pure white to yellowish white, yellowish white to greyish orange when dry.	Spines non-decurrent to subdecurrent; pure white when fresh, yellowish white when dry; 0.2–0.8 mm long.	(4.5)5–5.8(6) × (3.5)3.8–4.8(5) μm, av. 5.28 × 4.16 μm, Q = 1.27–1.31	In angiosperm forests; distributed in Hunan Province.	[2]
*H. cremeoalbum*	30–100 mm wide; white to cream white.	Spines adnate, non-decurrent; yellowish white, orange-white to pale orange; 2–5 mm long.	4–5.5 × 3.5–4.5 μm, av. 4.73 × 3.88 μm, Q = 1.1–1.42.	In broad-leaved or mixed forests; known from southern China and Japan.	[27] Present study
*H. flabellatum*	30–45 mm wide; yellowish white, pale yellow to greyish yellow, often with brownish orange scales.	Spines non-decurrent or subdecurrent; orange-white when fresh, greyish orange when dry; 0.6–2 mm long.	(7.8)8.5–9.5(10) × (6)6.5–7.8(8) μm, av. 9.07 × 7.04 μm, Q = 1.26–1.29	In mixed forests; distributed in Liaoning Province.	[2]
*H. flavidocanum*	20–30 mm wide; yellowish white or yellowish gray at center, whitish towards margin.	Spines non-decurrent or subdecurrent; orange-white when fresh, greyish orange when dry; 0.5–2 mm long,	(7)7.2–8.8(8.9) × (5.2) 5.5–6.5(6.8) μm, av. 7.75 × 6.01 μm, Q = 1.29–1.31	In mixed forests; distributed in Yunnan Province.	[2]
*H. jussii*	35–60 mm wide; very pale to medium orange ocher.	Spines somewhat decurrent; whitish to very pale ochraceous, later pale brownish ochraceous.	7.2–8 × 6.6–7.5 μm, av. 7.5 × 7 μm, Q = 1.03–1.18	In coniferous forests. Known from Finland and China.	[3,6]
*H. longibasidium*	10–15 mm wide; orange-white to greyish orange when fresh, greyish orange to brownish yellow when dry.	Spines non-decurrent or subdecurrent; orange-white to pale orange when fresh, greyish orange to brownish yellow when dry; 1–4 mm long.	(8)8.5–11(11.5) × (7.5)7.8–9.8(10) μm, av. 9.81 × 9.03 μm, Q = 1.09–1.13	In angiosperm forests; distributed in Hunan Province.	[2]
*H. longipes*	20–30 mm wide; orange, light orange, pastel red to reddish orange, changing to orange-white when bruised.	Spines adnate, non-decurrent; orange-white to pale orange; 1–3 mm long.	(7.5)8–10(10.5) × (6)7–8.5(9) μm, av. 9.32 × 7.63 μm, Q = 1.05–1.43	In *Quercus aquifolioides* and *Pinus densata* dominated forests; distributed in Yunnan Province	Present study
*H. melitosarx*	20–40 mm wide, reddish yellow, light yellow, light orange to orange.	Spines adnate; white, yellowish white to pale orange; 2–5 mm long.	8–11× 7–10 μm, av. 9.37 × 8.71 μm, Q = 1–1.28	In broad-leaved or mixed forests. Known from Asia, Europe, and North America.	[6] Present study
*H. microcarpum*	10–20 mm wide; white, yellowish white to orange-white, changing from brownish orange to brownish red when bruised.	Spines adnate to subdecurrent; white to yellowish white when fresh, grayish yellow to grayish orange when dry; 0.5–1.5 mm long.	5–6(6.5) × 5–5.5(6) μm, av. 5.78 × 5.26 μm, Q = 1–1.2	In mixed forests dominated; distributed in Guangdong Province	Present study
*H. minum*	10–25 mm wide; whitish, cream yellow to pale buff, changing reddish brown when bruised.	Spines adnate, non-decurrent; whitish, cream yellow to pale buff; 0.5–1.7 mm long.	4.5–5.5 × 3–4.5 μm, av. 4.8 × 3.8 μm, Q = 1.1–1.5	In mixed forest; Known from Japan and China.	[2,26]
*H. orientalbidum*	30–60 mm wide; white.	Spines adnate; white; 2–3 mm long.	4–5 × 3.5–4.5 μm, av. 4.65 × 3.96 μm, Q = 1.1–1.28	In broad-leaved forests. Known from China and Japan.	Present study
*H. pallidocroceum*	25–40 mm wide; orange-white to pale orange.	Spines non-decurrent; light yellow when fresh, orange-white to pale orange when dry; 1–5 mm long,	(7.5) 7.8–9.5(10) × (5.5)6–7.5(8) μm, av. 9.09 × 6.72 μm, Q = 1.32–1.35	In *Pinus* sp. and *Picea* sp. forests; distributed in Xinjiang Province.	[2]
*H. pallidomarginatum*	20–35 mm wide; orange-white to pale orange, with a light color zone towards center.	Spines decurrent; orange-white to pale orange when fresh, brownish orange when dry; 0.5–2 mm long,	(8) 8.2–9.8(10) × (6)6.5–7.8(8.2) μm, av. 8.75 × 6.99 μm, Q = 1.25–1.28	In angiosperm forests; distributed in Yunnan Province.	[2]
*H. pinicola*	25–60 mm wide; yellowish white, pale yellow to pale orange.	Spines adnate; orange-white, pale orange when fresh, pinkish white to pale red when dry; 1.5–3 mm long.	4–6 × 3.5–5 μm, av. 4.72 × 4.02 μm, Q = 1–1.25	In mixed forests; known from China and Japan.	[24] Present study
*H. sinorepandum*	40–120 mm wide; yellowish white, pale yellow, light yellow to light orange.	Spines adnate; white, yellowish white, pale yellow to pale orange; 3–7 mm long.	(7.5)8–9(10) × 6.5–7.5 μm, av. 8.16 × 7.12 μm, Q = 1–1.38	In broad-leaved and mixed forests; distributed in Yunnan Province.	Present study
*H. sphaericum*	20–35 mm wide; orange-white when moist, greyish orange to brownish orange when dry.	Spines non-decurrent to subdecurrent; white when fresh, brownish orange when dry; 0.5–3 mm long,	(7.5)8–8.8(9) × (6)6.5–7.5(8) μm, av. 8.36 × 6.94 μm, Q = 1.2–1.23	In angiosperm forest; distributed in Hunan Province.	[2]
*H. tangerinum*	10–50 mm wide; pale orange, light orange, orange to brownish orange when moist, greyish orange when dry.	Spines non-decurrent; orange-white when fresh, greyish orange when dry; 2–6 mm long.	(7)7.2–8.8(9) × (5.5) 5.8–7(7.5) μm, av. 8.11 × 6.19 μm, Q = 1.23–1.31	In angiosperm forest; distributed in Hunan Province	[2]
*H. tenuistipitum*	10–30 mm wide; yellow white to orange-white when moist, pale orange to greyish orange when dry.	Spines non-decurrent to subdecurrent; orange-white when fresh, light brown when dry; 1–3 mm long.	(6.5)6.8–7.2(7.5) × (5.2)5.5–6.5(6.8) μm, av. 7.08 × 6.09 μm, Q = 1.07–1.16	In angiosperm forest; distributed in Hunan Province.	[2]
*H. ventricosum*	28–35 mm wide; orange to brown.	Spines non-decurrent; orange-white when fresh, brownish orange when dry; 1–5 mm long.	(7.5)8.2–9(9.5) × (7)7.5–8.5(9) μm, av. 8.64 × 8.17 μm, Q = 1.05–1.09	In mixed forests; distributed in Liaoning Province.	[2]
*H. vesterholtii*	10–30(–50) mm wide, ocher to light ocher.	Spines non-decurrent to slightly decurrent; pale ocher; 1–1.7 mm long.	(7)8–9(9.5) × 6–7.5(8) μm, av. 8.2–8.7 × 6.4–6.8 μm; Q = 1.27–1.3;	In Abies alba and Fagus sylvatica forest. Known from China and France.	[3,29]

## Data Availability

The datasets presented in this study can be found in online repositories. The names of the repository/repositories and accession number(s) can be found below: https://nmdc.cn/fungalnames/, and https://www.ncbi.nlm.nih.gov/genbank/ (accessed on 8 December 2023).

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
