# Peer review of "A Contribution to the Phylogeny and Taxonomy of Hydnum (Cantharellales, Basidiomycota) from China"

_jof, 2024, doi:10.3390/jof10020098_

Round 1

Reviewer 1 Report

Comments and Suggestions for Authors

Overall, this was an interesting description of Hydnum spp. in China with the description of three new species. The manuscript format seems to align with other article describing new species of mushrooms.

My concerns with the manuscript are the phylogenetic trees. While I was able to read the phylogenies, they still were not very clear. I think the clarity of the trees can still be improved to help reads. Also, there was some moderate text editing for clarity. Higher resolution figures are needed for readers to assess the results.

Comments on the Quality of English Language

Overall the English is good. There are some minor grammatical errors. 

The phrase 'firstly described in China' is a bit confusing especially in the abstract. It seems like all the species mentioned were originally described in China, but this was not the case. 

Author Response

Dear reviewer,

Thank you for your comments concerning our manuscript. We have revised our MS carefully which we hope meet with approval.

All  the changes have been marked in the text in the revised file.

Once again, thank you very much for your comments and suggestions.

Sincerely,

Ming Zhang

Reviewer 2 Report

Comments and Suggestions for Authors

Currently, the identification of fungal species requires comprehensive taxonomy, which involves the study of morphological and molecular characteristics, as well as the environmental factors in which fungi develop.

In this manuscript, the description of the macroscopic, microscopic and molecular characteristics of the identified species is made, but the description of the environment in which they develop is secondary.

It is suggested that this information be further investigated so that the identification is more robust.

On the other hand, a table must be included with the comparison of the characteristics presented by all the species, as support for the new species and those previously described.

As the information is presented it is difficult to understand the distinctive characteristics of the species. It is also recommended to include photographs of the most important microscopic structures for each species.

Author Response

Dear reviewer,

Thank you for your comments concerning our manuscript. We have revised our MS carefully which we hope meet with your approval.

All your comments have been answered in the manuscript and here, and marked in revised format in the revised file. 

Comment 1: In this manuscript, the description of the macroscopic, microscopic and molecular characteristics of the identified species is made, but the description of the environment in which they develop is secondary.

Response: We are not quite sure what environmental factors the reviewer refers to. But the ecological information relevant to each species has been provided under “habitat and distribution”.

Comment 2: On the other hand, a table must be included with the comparison of the characteristics presented by all the species, as support for the new species and those previously described.

Response: It is a good suggestion to add a feature comparison table, but it is impossible and useless to list all known species, and most Hydnum species have restricted distribution areas, thus, a table of morphological comparison of Hydnum species in China has been added.

Comment 3: As the information is presented it is difficult to understand the distinctive characteristics of the species. It is also recommended to include photographs of the most important microscopic structures for each species.

Response: The line drawing of the most important microscopic structures for each species have been provided in the text, and there is no need to repeat the microscopic photos.

Once again, thank you very much for your comments and suggestions.

Sincerely,

Ming Zhang

Round 2

Reviewer 1 Report

Comments and Suggestions for Authors

The revised manuscript is sufficient. The phylogenetic trees included in the revised version are still of poor quality. Hopefully, higher quality figures like those provided after my initial review will be used for the published version. 

Comments on the Quality of English Language

I still recommend getting someone to go through the manuscript to edit the English. As is, the manuscript is understandable, but a little editing will make it a little easier for readers.

Author Response

Dear Reviewer, 

Thank you very much for your comments. We have revised our manuscript according to your comments, the changes have been marked in the revised file, and now the figures of phylogenetic tree is clear. We hope our revised file can meet with your approval.

Best wishes,

Ming Zhang

Reviewer 2 Report

Comments and Suggestions for Authors The article is relevant for the description of 3 new species of Hydnum for China. The attached document presents some comments and questions that are required to strengthen the manuscript. When describing new species, the information on their morphology, ecology, and molecular data obtained must be reinforced. In several cases this information is cursory and may remain ambiguous. In general, the number of specimens used for characterization is few, 2 or 3 at most, which means that the genetic diversity is not shown in detail. It is recommended to include a greater number of specimens in each of the species. The figures of the phylogenetic trees must be improved, their visualization is impossible. Improve image names about microscopic structures. Integrate a taxonomic key of the species represented.    

Author Response

Dear Reviewer, 

Thank you very much for your comments. We have revised our manuscript according to your comments, the changes have been marked in the revised file, and were also listed below. We hope our revised file can meet with your approval.

Best wishes,

Ming Zhang

Comments 1: Identify the name of the marked structure...does it refer to the granular substance? also mark the clamp connections

Response 1: Yes, the marked substance on the basidioles refers to the basidioles covered with yellowish granular substance; and the this structure marked in the pileipellis refers to the clamp connections. They all were indicated in the figure notes.

Comments 2: The characterization of the habitat and distribution should be more specific and mention the tree species present, since currently they are characteristics that may have taxonomic importance to distinguish the species.

Response 2: This is a good common, and we have added relevant ecological information and possible host plants in the text.

Comments 3: In addition to the table, it is important to integrate a taxonomic key that helps distinguish the most relevant morphological characteristics for each species. On the other hand, base on molecular techniques for the characterization of the species.

Response 3: A key to species of Hydnum in China was added in the text.